# A hydrophobic gate in the inner pore helix is the major determinant of inactivation in mechanosensitive Piezo channels

Wang Zheng[1], Elena O Gracheva[1,2,3], Sviatoslav N Bagriantsev[1]*

[1]Department of Cellular and Molecular Physiology, Yale University School of Medicine, New Haven, United States; [2]Department of Neuroscience, Yale University School of Medicine, New Haven, United States; [3]Program in Cellular Neuroscience, Neurodegeneration and Repair, Yale University School of Medicine, New Haven, United States

**Abstract** Piezo1 and Piezo2 belong to a family of mechanically-activated ion channels implicated in a wide range of physiological processes. Mechanical stimulation triggers Piezo channels to open, but their characteristic fast inactivation process results in rapid closure. Several disease-causing mutations in Piezo1 alter the rate of inactivation, highlighting the importance of inactivation to the normal function of this channel. However, despite the structural identification of two physical constrictions within the closed pore, the mechanism of inactivation remains unknown. Here we identify a functionally conserved inactivation gate in the pore-lining inner helix of mouse Piezo1 and Piezo2 that is distinct from the two constrictions. We show that this gate controls the majority of Piezo1 inactivation via a hydrophobic mechanism and that one of the physical constrictions acts as a secondary gate. Our results suggest that, unlike other rapidly inactivating ion channels, a hydrophobic barrier gives rise to fast inactivation in Piezo channels.
DOI: https://doi.org/10.7554/eLife.44003.001

*For correspondence:
slav.bagriantsev@yale.edu

Competing interests: The authors declare that no competing interests exist.

## Introduction

The mechanically gated ion channels, Piezo1 and Piezo2, are critical for a broad range of processes involving mechanotransduction in both neuronal and non-neuronal cells (*Coste et al., 2010*; *Coste et al., 2012*; *Ranade et al., 2015*; *Wu et al., 2017a*). Piezo1 has been implicated in blood pressure regulation, vascular development, arterial remodeling, neural stem cell fate determination and endothelium homeostasis (*Rode et al., 2017*; *Gudipaty et al., 2017*; *Wang et al., 2016*; *Koser et al., 2016*; *Retailleau et al., 2015*; *Blumenthal et al., 2014*; *Pathak et al., 2014*; *Li et al., 2014*; *Ranade et al., 2014a*; *McHugh et al., 2012*; *Zeng et al., 2018*; *Szczot et al., 2017*; *Del Mármol et al., 2018*). Piezo2 is expressed in somatosensory neurons where it plays a major role in the transduction of gentle and painful touch, proprioception, airway stretch, lung inflation and neuronal migration (*Nonomura et al., 2017*; *Woo et al., 2015*; *Ranade et al., 2014b*; *Ikeda et al., 2014*; *Woo et al., 2014*; *Maksimovic et al., 2014*). Malfunctions in Piezo1 or Piezo2 are associated with a number of human diseases that involve mechanotransduction, including xerocytosis, arthrogryposis, lymphedema and hyperalgesia (*Murthy et al., 2018*; *Szczot et al., 2018*; *Haliloglu et al., 2017*; *Alisch et al., 2017*; *Fotiou et al., 2015*; *Cahalan et al., 2015*; *McMillin et al., 2014*; *Andolfo et al., 2013*; *Glogowska et al., 2017*; *Chesler et al., 2016*; *Zarychanski et al., 2012*). Mechanical stimulation of Piezo channels gives rise to a mechanically-activated (MA) current, which quickly decays due to fast inactivation (*Lewis et al., 2017*; *Gottlieb et al., 2012*). Disease-linked

mutations in Piezo1 and Piezo2 specifically affect this inactivation process, suggesting that the normal timing of MA current decay is important for animal physiology (*Wu et al., 2017a*). In addition, a prolongation of Piezo2 inactivation in somatosensory neurons of tactile-specialist birds suggests that inactivation is involved in the modulation of complex behaviors (*Schneider et al., 2017*; *Anderson et al., 2017*; *Schneider et al., 2014*). Inactivation is significantly affected by the known modulators of Piezo1: Yoda1 and Jedi1/2 (*Lacroix et al., 2018*; *Wang et al., 2018*; *Evans et al., 2018*; *Syeda et al., 2015*). Yet, despite its significance for channel function, physiology and pathophysiology, the mechanism of Piezo inactivation remains unknown.

Functional Piezo channels are homo-trimers that adopt a unique propeller-like architecture comprising a central C-terminal ion-conducting pore and three peripheral N-terminal blades (*Figure 1A*) (*Guo and MacKinnon, 2017*; *Saotome et al., 2018*; *Zhao et al., 2018*). Each blade is composed of 36 transmembrane (TM) segments and is thought to contribute to sensing tension in the membrane (*Guo and MacKinnon, 2017*; *Haselwandter and MacKinnon, 2018*). The pore region, which contains an outer pore helix (OH), an inner pore helix (IH), an extracellular cap domain and an intracellular C-terminal domain (CTD), is responsible for ion conduction. The ion permeation pathway is lined by the IH within the membrane and is surrounded by the CTD as it continues into the cytoplasm. All three cryo-electron microscopy (cryo-EM) structures of Piezo1 indicate the presence of two physical constrictions in the CTD: one formed by residues M2493/F2494 (MF constriction) and the other by residues P2536/E2537 (PE constriction) (*Figure 1B and C*) (*Zhao et al., 2018*; *Saotome et al., 2018*; *Guo and MacKinnon, 2017*). These constrictions define minimum pore diameters of 6 Å and 4 Å, respectively, thus the structures are assumed to represent a closed state.

Here, we combine electrophysiology and mutagenesis to investigate the mechanism of inactivation in Piezo1 and Piezo2. We show that the major inactivation element comprises two conserved hydrophobic residues, located above the MF and PE constrictions, in the middle portion of the inner helix. The constrictions evident in Piezo1 structures play moderate roles in Piezo1 inactivation. Our results suggest that Piezo1 inactivation is accomplished by at least two gates, one of which acts as a hydrophobic barrier.

## Results

### Physical constrictions in the CTD play only moderate roles in Piezo1 inactivation

We first sought to determine whether the MF and PE constrictions evident in the CTD of Piezo1 structures contribute to inactivation of Piezo1-mediated MA current. To test this, we introduced mutations at the M2493/F2494 site and assessed the rate of MA current inactivation in HEK293-$^{PIEZO1-/-}$ (HEK293T$^{\Delta P1}$) cells (*Dubin et al., 2017*; *Lukacs et al., 2015*) in response to a 300 ms mechanical indentation with a glass probe. Overexpression of wild-type (WT) mouse Piezo1 in HEK293T$^{\Delta P1}$ cells produced robust MA currents with fast inactivation kinetics (time constant of inactivation ($\tau_{inact}$) = 11.9 ± 0.6 ms) (*Figure 1D*). Systematic amino acid substitutions at the M2493/F2494 site to hydrophilic or hydrophobic residues had either no effect on $\tau_{inact}$ (MF/SS, $\tau_{inact}$ = 13.3 ± 1.1 ms) or prolonged $\tau_{inact}$ by 1.6–2.7 fold (MF/QQ, NN, TT, GG, AA, VV, LL, II, WW, average $\tau_{inact}$ = 19.4–31.9 ms) (*Figure 1D*). These data reveal that the MF site only moderately contributes to Piezo1 inactivation. Moreover, even though the MF constriction is formed by hydrophobic residues (*Figure 1C*), we found no correlation between the rate of Piezo1 inactivation and hydrophobicity at this site.

Next, we investigated the P2536/E2537 constriction, which is located more cytoplasmically than the MF constriction and forms a smaller diameter aperture (*Figure 1B and C*). Mutating P2536 and E2537 to glycines resulted in substantially reduced peak MA currents with only slightly prolonged inactivation ($\tau_{inact}$ = 17.6 ± 0.8 ms) (*Figure 1E–G*). These data suggest that the PE constriction is unlikely to be involved in Piezo1 inactivation. Instead, we found that the PE/GG mutation dramatically accelerated deactivation kinetics of a Piezo1 mutant (see below and *Figure 1—figure supplement 1*). Together, these data show that the physical constructions at the MF and PE sites in the CTD are important for channel function, but only moderately affect Piezo1 inactivation, suggesting that the main inactivation mechanism is located elsewhere in the channel.

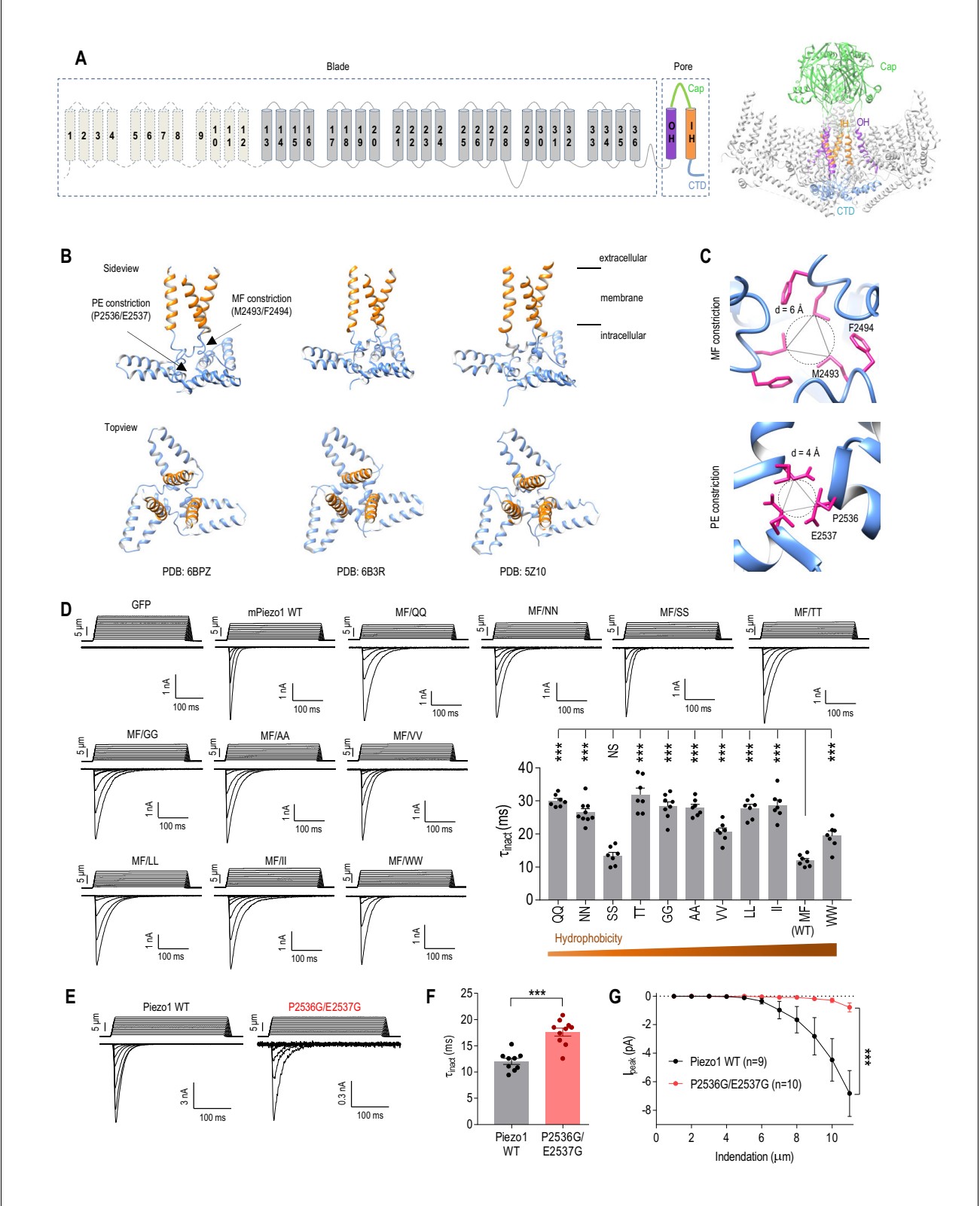

**Figure 1.** Physical constrictions in the CTD play only moderate roles in Piezo1 inactivation. (**A**) Topology diagram and cryo-EM structure of Piezo1 (PDB: 6BPZ). IH, inner pore helix; OH, outer pore helix. (**B**) Side view and top view of the Piezo1 pore region from three cryo-EM structures showing the location of the MF and PE constrictions. (**C**) Top view close-up of the MF and PE constrictions in Piezo1 (PDB: 6BPZ). (**D**) Representative whole-cell MA current traces and quantification of MA current inactivation rate ($\tau_{inact}$) in HEK293T$^{\Delta P1}$ cells expressing Piezo1 with mutations at the M2493 F2494 (MF)

*Figure 1 continued on next page*

*Figure 1 continued*

site (n = 7–9 cells). E$_{hold}$ = −80 mV. ***p<0.001; NS, not significant, p>0.05, one-way ANOVA with Holm-Sidak's correction. (**E and F**) Representative whole-cell MA current traces and quantification of MA current inactivation for WT Piezo1 and P2536G/E2537G mutant. **p<0.001, unpaired t-test. (**G**) Quantification of peak MA current amplitude (I$_{peak}$) at different indentation depths for WT Piezo1 and P2536G/E2537G mutant. ***p<0.001, two-way ANOVA. Data are mean ± SEM.

DOI: https://doi.org/10.7554/eLife.44003.002

The following source data and figure supplements are available for figure 1:

**Source data 1.** Electrophysiological analysis of Piezo1 CTD mutants.

DOI: https://doi.org/10.7554/eLife.44003.005

**Figure supplement 1.** Mutations at the Piezo1 PE site accelerate deactivation of MA current.

DOI: https://doi.org/10.7554/eLife.44003.003

**Figure supplement 1—source data 1.** Electrophysiological analysis of Piezo1 PE site mutants.

DOI: https://doi.org/10.7554/eLife.44003.004

## The pore-lining inner helix plays a major role in Piezo1 inactivation

In search of the main structural element(s) of Piezo1 inactivation, we investigated the pore-lining inner helix (IH). We noticed that the middle portion of IH is lined with pore-facing hydrophobic residues (L2469, I2473, V2476 and F2480), two of which are contained within a cluster of conserved amino acids ($_{2473}$IVLVV$_{2477}$, *Figure 2A*). To examine whether these hydrophobic residues play a role in Piezo1 inactivation, we replaced each of them with a hydrophilic serine. We found that serine substitutions at L2475 and V2476, but not at other positions, significantly prolonged inactivation (L2475S, τ$_{inact}$ = 62.2 ± 2.1 ms; V2476S, τ$_{inact}$ = 46.8 ± 1.7 ms) (*Figure 2B*). Combining the two mutations had a cumulative effect, resulting in an almost ten-fold increase in τ$_{inact}$ (L2475S/V2476S, τ$_{inact}$ = 103.3 ± 2.9 ms). These data indicate that the L2475/V2476 (LV) site forms part of the inactivation mechanism of Piezo1. Interestingly, the LV/SS mutant exhibited a persistent current after removal of the mechanical stimulus (*Figure 2B*). The decay of the persistent current reflects deactivation of Piezo1 (*Wu et al., 2016*), which can be substantially accelerated by the P2536G/E2537G double mutation in the PE constriction (*Figure 1—figure supplement 1*). This supports the idea that the PE constriction could be involved in Piezo1 deactivation, in contrast to the inner helix LV site, which mediates inactivation.

Next, we asked whether mutations at L2475 and V2476 affect inactivation specifically. We found that individual or combined serine substitutions at these sites had no effect on whole-cell MA current amplitude (*Figure 2C*), apparent threshold of mechanical activation (*Figure 2D*), MA current rise time (*Figure 2E*), or rectification and relative ionic selectivity (*Figure 2F and G*). Similar to WT Piezo1, the inactivation rate of the L2475S and V2476S mutants slowed with depolarization (*Figure 2H*), demonstrating that the mutations did not affect the voltage dependence of inactivation (*Coste et al., 2010; Moroni et al., 2018; Wu et al., 2017b*). Furthermore, the mutations did not affect basal current in the absence of mechanical stimulation, supporting the conclusion that these amino acids do not contribute to channel activation (*Figure 2—figure supplement 1*). Taken together, these results show that residues L2475 and V2476 are specifically involved in Piezo1 inactivation.

## The hydrophobicity of L2475 and V2476 determines the rate of Piezo1 inactivation

Following our observation that the LV site forms part of a hydrophobic cluster in the pore-lining IH (*Figure 2A*), we hypothesized that the hydrophobicity of these residues determines Piezo1 inactivation. Strikingly, we found a strong correlation between hydrophobicity and the rate of Piezo1 inactivation at both positions. Mutating L2475 to the highly hydrophilic Q or N led to a substantial ~11 fold increase in τ$_{inact}$ (L/Q, τ$_{inact}$ = 124.5 ± 4.4 ms; L/N, τ$_{inact}$ = 112.7 ± 5.4 ms) (*Figure 3A*). Mutations to ether serine or threonine produced a significant, but moderate increase (L/S, τ$_{inact}$ = 62.2 ± 2.1 ms; L/T, τ$_{inact}$ = 25.9 ± 1.8 ms). Bulky hydrophobic amino acid substitutions, on the other hand, led to either similar or faster inactivation compared to WT Piezo1 (L/V, τ$_{inact}$ = 2.8 ± 0.3 ms; L/I, τ$_{inact}$ = 2.8 ± 0.2 ms; L/F, τ$_{inact}$ = 10.2 ± 0.4 ms) (*Figure 3A*). The small hydrophobic G or A substitutions at L2475 resulted in a smaller increase in τ$_{inact}$ compared to the effects of large hydrophilic Q or N

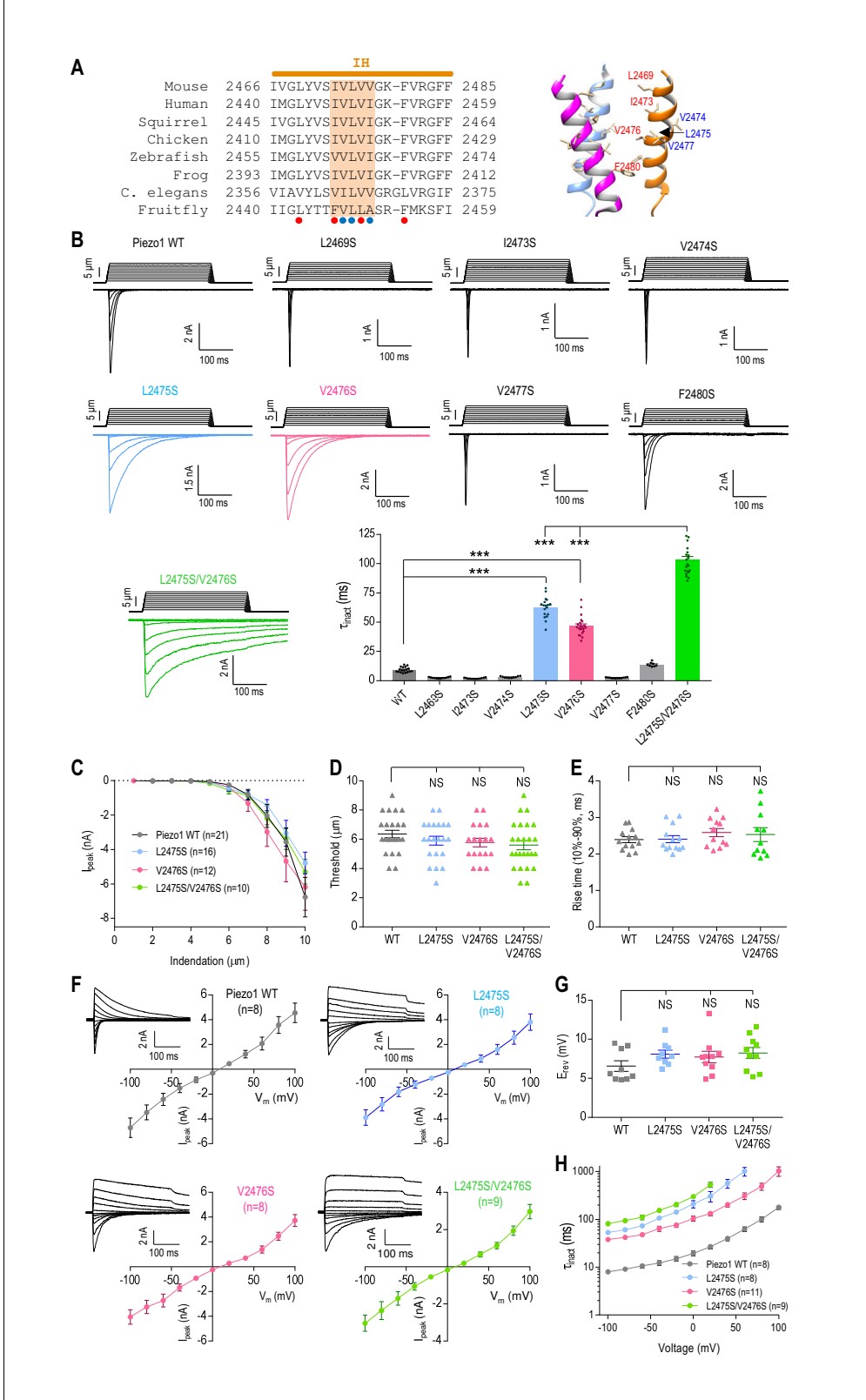

**Figure 2.** The pore-lining inner helix plays a major role in Piezo1 inactivation. (A) Left panel, amino acid sequence alignment of the Piezo1 inner helix (IH) from different species. A cluster of five conserved hydrophobic residues in the middle are highlighted. Red and blue dots indicate hydrophobic residues facing and pointing away from the pore, respectively. Right panel, cryo-EM structure of the Piezo1 inner helix (PDB: 6BPZ) showing the hydrophobic residues in the left panel. (B) Representative whole-cell MA current traces and quantification of MA current inactivation rate ($\tau_{inact}$) in

*Figure 2 continued on next page*

*Figure 2 continued*

HEK293T$^{\Delta P1}$ cells expressing Piezo1 with mutations in the hydrophobic cluster in the inner helix (n = 8–22 cells). $E_{hold}$ = −80 mV. ***p<0.001; NS, not significant, p>0.05, one-way ANOVA with Dunnet's correction. (C–E) Quantification of peak MA current amplitude ($I_{peak}$) at different indentation depths (C), apparent indentation threshold of MA current activation (D) and MA current rise time (E) for WT and mutant Piezo1. NS, not significant, p>0.05, one-way ANOVA with Dunnet's correction. (F) Peak MA current-voltage relationship in response to mechanical indentation at 9 μm for WT Piezo1 or indicated mutants. Insets show representative traces of whole-cell MA currents evoked at $E_{hold}$ ranging from −100 mV to +100 mV, in 20 mV increments. (G) Quantification of the reversal potential ($E_{rev}$) from current-voltage plots in (F). NS, not significant, p>0.05, one-way ANOVA with Dunnet's correction. (H) Quantification of MA current inactivation rate for WT or mutant Piezo1 at different voltages. Data are mean ± SEM.
DOI: https://doi.org/10.7554/eLife.44003.006

The following source data and figure supplements are available for figure 2:

**Source data 1.** Electrophysiological analysis of Piezo1 IH mutants.
DOI: https://doi.org/10.7554/eLife.44003.009
**Figure supplement 1.** Mutations that prolong inactivation in Piezo1 do not affect basal current.
DOI: https://doi.org/10.7554/eLife.44003.007
**Figure supplement 1—source data 1.** Quantification of basal current in Piezo1 mutants.
DOI: https://doi.org/10.7554/eLife.44003.008

substitutions (L/G, $\tau_{inact}$ = 40.2 ± 1.4 ms; L/A, $\tau_{inact}$ = 22.1 ± 1.4 ms), lending support to the idea that hydrophobicity is the main factor determining Piezo1 inactivation at L2475 (*Figure 3A*). We also found a similar correlation between hydrophobicity at the V2476 position and inactivation rate (*Figure 3B*), suggesting that both residues contribute to Piezo1 inactivation via a similar mechanism. Importantly, the isosteric polar substitutions L2475N and V2476T, which presumably decrease hydrophobicity without affecting the size of the pore, both slowed Piezo1 inactivation. This underscores the importance of hydrophobicity, rather than pore size, in determining inactivation at these two positions. We therefore propose that L2475 and V2476 together form a hydrophobic inactivation gate in Piezo1.

## Mutation of the inner helix and MF constriction eliminates Piezo1 inactivation

If the putative hydrophobic gate formed by the LV site is the only inactivation gate in Piezo1, then replacement of both residues with highly hydrophilic glutamines should lead to a complete loss of inactivation. Because long inactivation times render the use of $\tau_{inact}$ as a measure of current decay inefficient, we tested this hypothesis by measuring the fraction of remaining MA current during 300 ms mechanical stimuli compared to peak current ($I_{remaining}/I_{peak}$). We found that the LV/QQ double mutant exhibited only a marginal prolongation of inactivation compared to the single substitutions ($I_{remaining}/I_{peak}$ at 300 ms, mean ± SEM: WT, 0.0058 ± 0.0007; L2475Q, 0.41 ± 0.03; V2476Q, 0.19 ± 0.03; LV/QQ, 0.49 ± 0.03) (*Figure 4A and B*). Thus, even though the majority of inactivation was eliminated in the LV/QQ mutant, the channel still exhibited some current decay, suggesting that another gate contributes to inactivation. Because Piezo1 inactivation is partially determined by the MF constriction in the CTD (*Figure 1D*), we introduced the MF/QQ mutations into the LV/QQ channel. Strikingly, the resultant quadruple mutant (LV/QQ-MF/QQ) showed a complete loss of inactivation ($I_{remaining}/I_{peak}$ = 0.89 ± 0.03 at 300 ms) (*Figure 4A and B*). We also consistently observed complete elimination of inactivation in Piezo1 by high speed pressure clamp in the cell-attached configuration, demonstrating that this result is independent of the method of mechanical stimulation (*Figure 4C*). Thus, our data suggest that the MF constriction in the CTD could act in concert with the inner helix hydrophobic LV gate to produce fast inactivation of Piezo1. Collectively, these data reveal that the two putative inactivation gates are sufficient to account for the inactivation of Piezo1 during mechanical stimulation.

## The putative inner helix inactivation gate is functionally conserved in Piezo2

The L2475 and V2476 residues are conserved in the Piezo1 homologue, Piezo2 (L2750 and V2751, respectively) (*Figure 5A*). We therefore sought to determine whether these hydrophobic residues are also involved in Piezo2 inactivation. Substituting L2750 or V2751 with hydrophilic serine significantly prolonged inactivation (WT, $\tau_{inact}$ = 2.5 ± 0.1 ms; L2750S, $\tau_{inact}$ = 8.3 ± 0.5 ms; V2751A, $\tau_{inact}$

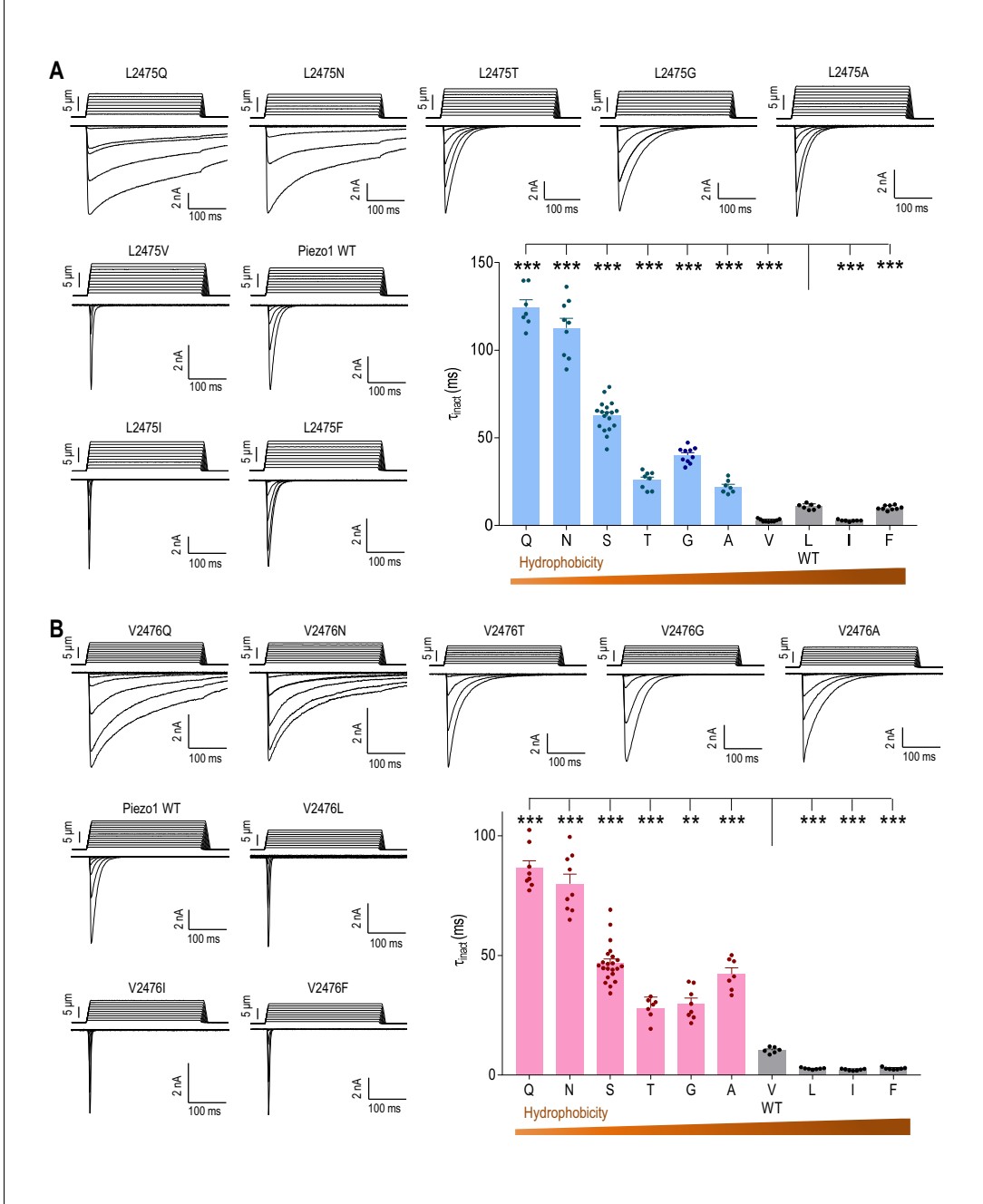

**Figure 3.** The hydrophobicity of L2475 and V2476 determines the rate of Piezo1 inactivation. (*A and B*) Representative whole-cell MA current traces and quantification of MA current inactivation rate ($\tau_{inact}$) in HEK293T$^{\Delta P1}$ cells expressing Piezo1 with indicated mutations of variable hydrophobicity at L2475 (*A*, n = 7–18 cells) and V2476 (*B*, n = 6–22 cells). $E_{hold}$ = −80 mV. **p<0.01, ***p<0.001, one-way ANOVA with Holm-Sidak's correction. Data are mean ± SEM.

DOI: https://doi.org/10.7554/eLife.44003.010

The following source data is available for figure 3:

**Source data 1.** Electrophysiological analysis of Piezo1 L2475 and V2476 mutants.
DOI: https://doi.org/10.7554/eLife.44003.011

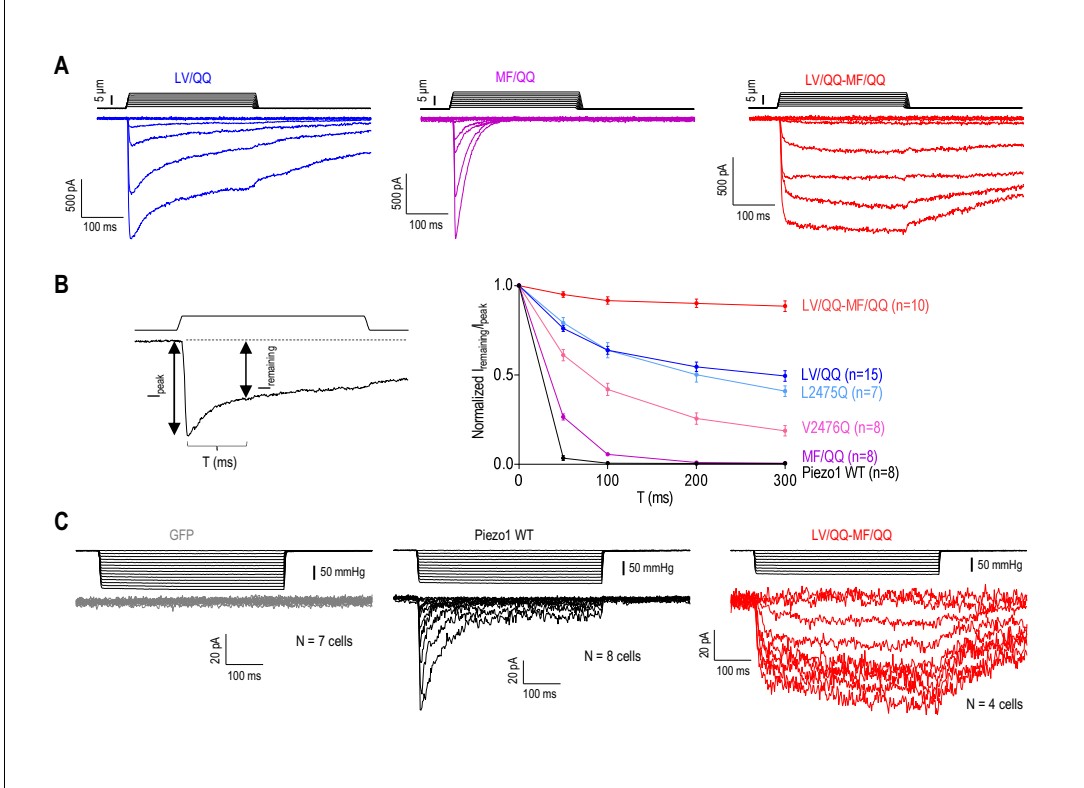

**Figure 4.** Mutation of the inner helix and MF constriction eliminates Piezo1 inactivation. (**A**) Representative whole-cell MA current traces from HEK293T$^{\Delta P1}$ cells expressing Piezo1 with glutamine mutations in the putative hydrophobic gate (L2475/V2476, LV), or the MF constriction (M2493/F2494, MF). $E_{hold}$ = −80 mV. (**B**) Left panel, an example trace of Piezo1 MA current illustrating the measurement of the ratio of remaining MA current amplitude ($I_{remaining}$) to peak ($I_{peak}$) at different time points during current decay. Right panel, quantification of $I_{remaining}/I_{peak}$ for WT or mutant Piezo1. Data are mean ± SEM. (**C**) Representative cell-attached MA current traces induced by high-speed pressure clamp via application of a negative pipette pressure in HEK293T$^{\Delta P1}$ cells expressing GFP (negative control), WT or mutant Piezo1. $E_{hold}$ = −80 mV.

DOI: https://doi.org/10.7554/eLife.44003.012

The following source data is available for figure 4:

**Source data 1.** Quantification of current decay in Piezo1 mutants.
DOI: https://doi.org/10.7554/eLife.44003.013

= 14.2 ± 1.4 ms) (*Figure 5B and C*). The double mutants LV/SS and LV/QQ did not result in functional channels. The effects of these serine substations were specific to inactivation and did not affect whole-cell MA current amplitude (*Figure 5D*), apparent activation threshold (*Figure 5E*), current rise time (*Figure 5F*), relative ion permeability (*Figure 5G–I*), or voltage dependence of inactivation (*Figure 5J*). These data suggest that the LV site in Piezo2 is specifically involved in inactivation, and that the putative inactivation gate in the inner helix is functionally conserved among Piezo channels. We also investigated the region in Piezo2 that is homologous to the secondary MF inactivation gate in Piezo1. In contrast to Piezo1, substituting M2767 and F2768 (homologous to M2493 and F2494 in Piezo1) with glutamines did not affect inactivation (MF/QQ, $\tau_{inact}$ = 2.7 ± 0.2 ms) (*Figure 5B and C*). These results show that, even though Piezo1 and Piezo2 share common elements of inactivation, their mechanisms are not identical and involve components specific to each channel.

## Discussion

The duration of Piezo-mediated mechanosensitive currents are important for the physiology of various types of neuronal and non-neuronal cells. Indeed, several disease-linked mutations in Piezo1 slow the inactivation of MA currents, but the molecular mechanism of this process remains elusive. We set out to investigate the molecular basis of Piezo channel inactivation and identified two

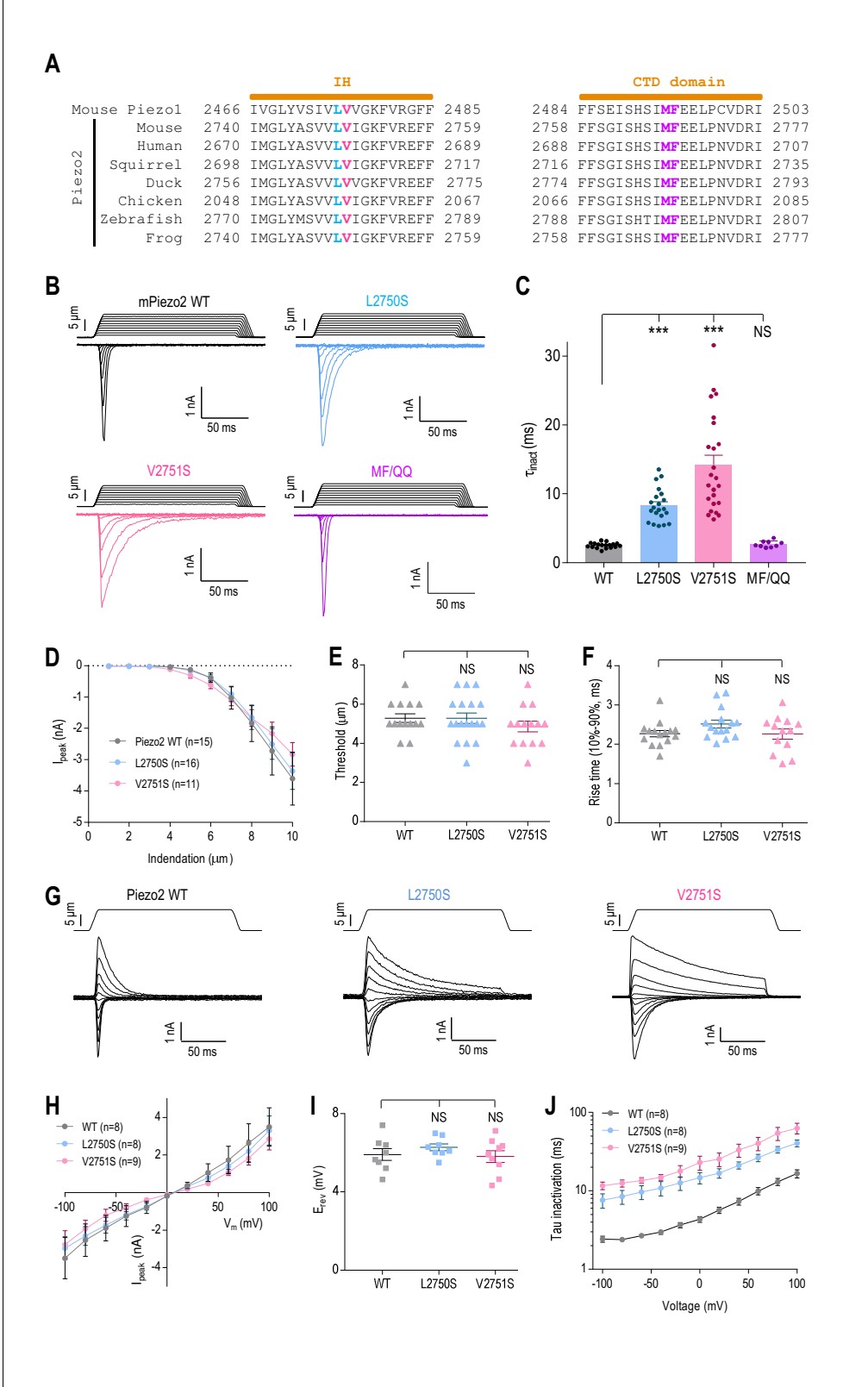

**Figure 5.** The putative inner helix inactivation gate is functionally conserved in Piezo2. (**A**) Amino acid sequence alignments of the IH and part of CTD between mouse Piezo1 and Piezo2 orthologues from indicated species. The conserved L2475 and V2476 residues in the IH are highlighted in blue and red; M2493 and F2494 in the CTD are highlighted purple. (**B and C**) Representative whole-cell MA current traces of WT and mutant Piezo2 (**B**), and

*Figure 5 continued on next page*

*Figure 5 continued*

quantification of MA current inactivation constant ($\tau_{inact}$) in HEK293T$^{\Delta P1}$ cells (**C**, n = 9–24 cells). E$_{hold}$ = −80 mV. Data are mean ± SEM. **p<0.001; NS, not significant, one-way ANOVA with Dunnett's correction. (**D–F**) Quantification of peak MA current amplitude (I$_{peak}$) at different indentation depths (**D**), apparent indentation threshold of MA current activation (**E**) and MA current rise time (**F**) for WT and mutant Piezo2 in HEK293T$^{\Delta P1}$ cells. E$_{hold}$ = −80 mV. NS, not significant, p>0.05, one-way ANOVA with Dunnet's correction. (**G and H**) Representative current traces (**G**) and quantification of peak MA current-voltage relationship (**H**) in response to mechanical indentation at 9 μm for WT or mutant Piezo2, evoked at E$_{hold}$ ranging from −100 mV to +100 mV, in 20 mV increments. (**I**) Quantification of the reversal potential (E$_{rev}$) from current-voltage plots in (**H**). NS, not significant, p>0.05, one-way ANOVA with Dunnet's correction. (**J**) Quantification of MA current inactivation rate for WT or mutant Piezo2 in response to a 9 μm indentation at different voltages. Data are mean ±SEM.

DOI: https://doi.org/10.7554/eLife.44003.014

The following source data is available for figure 5:

**Source data 1.** Electrophysiological analysis of Piezo2 mutants.
DOI: https://doi.org/10.7554/eLife.44003.015

---

conserved hydrophobic residues in the inner helix (L2475 and V2476) as the major determinants of inactivation in Piezo1. We also found that mutation of a physical constriction in the cytoplasmic end of the pore – the MF constriction formed by residues M2493 and F2494 in the CTD (*Zhao et al., 2018*; *Saotome et al., 2018*; *Guo and MacKinnon, 2017*) – abolishes all remaining inactivation in LV mutants. Collectively, our data lead us to conclude that the two residues at the LV site form a hydrophobic inactivation gate, which contributes to the majority of MA current decay (primary inactivation gate), and that the MF constriction acts as a secondary inactivation gate in Piezo1.

To form a hydrophobic inactivation gate, both L2475 and V2476 residues would have to face the pore in the inactivated state. Interestingly, however, the cryo-EM structures of Piezo1 in a closed state (*Zhao et al., 2018*; *Saotome et al., 2018*; *Guo and MacKinnon, 2017*) reveal that only the V2476 residue faces the pore, and that the L2475 residue points away from the pore (*Figure 6A*). We therefore propose that Piezo1 inactivation might involve a twisting motion of the IH to allow both L2475 and V2476 residues to face the ion-conducting pore (*Figure 6B*). The physical diameter of the closed pore at V2476 is 10 Å. For a hydrophobic gate to form an energetic barrier to ionic flow, its pore diameter should be less than 6 Å (*Zheng et al., 2018b*). Thus, in addition to the twisting motion, we also expect the IH to undergo a motion that leads to pore constriction (*Figure 6B*). The combined twisting and constricting motions of the IH may allow L2475 and V2476 to close the pore by forming a hydrophobic barrier, rather than by physically occluding the pore, but this hypothetical mechanism remains to be tested by obtaining structures in different conformations.

Hydrophobic gating was initially proposed after observing unusual liquid-vapor transitions of water molecules within model hydrophobic nanopores during molecular dynamics simulations (*Beckstein and Sansom, 2003*; *Hummer et al., 2001*). The transient vapor states are devoid of water within the pore, causing an energetic barrier to ion permeation. Thus, a hydrophobic gate stops the flow of ions even when the physical pore size is bigger than that of the ion (*Rao et al., 2018*). Over the past decade, evidence has accumulated to suggest that hydrophobic gating is widely present in ion channels (*Rao et al., 2018*; *Aryal et al., 2015*). In most cases, hydrophobic gates act as activation gates. For example, even though a number of TRP channels, including TRPV1, have a gating mechanism similar to that found in voltage-gated potassium channels (*Salazar et al., 2009*), others, such as TRPP3 and TRPP2 contain a hydrophobic activation gate in the cytoplasmic pore-lining S6 helix, which was revealed by both electrophysiological (*Zheng et al., 2018b*; *Zheng et al., 2018a*) and structural studies (*Cheng, 2018*). The bacterial mechanosensitive ion channels, MscS and MscL, also contain a hydrophobic activation gate (*Beckstein et al., 2003*). Our data suggest that the putative hydrophobic gate in Piezo1 seems to act as a major inactivation gate. Importantly, serine mutations at L2475 and V2476 specifically modulate Piezo1 inactivation without affecting other functional properties of the channel, including peak current amplitude and activation threshold. We also did not detect a change in MA and current rise time, even though a small change could avoid detection due to limitations imposed by the velocity of the mechanical probe. These results indicate that activation and inactivation gates are formed by separate structural elements in

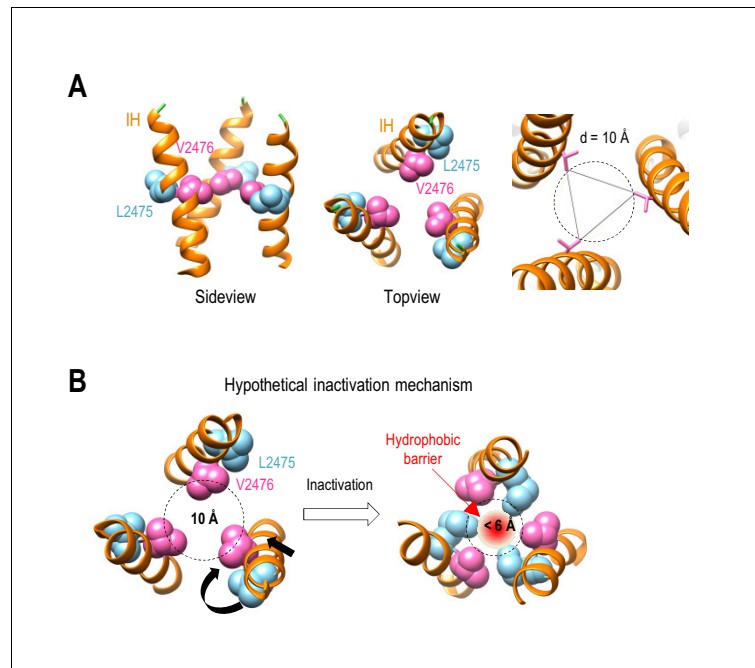

**Figure 6.** Hypothetical inactivation mechanism of Piezo1. (**A**) Left and middle panels, the side view and top view of a portion of Piezo1 inner helix (PDB: 6BPZ) showing the orientations of L2475 and V2476 residues with respect to the ion permeation pore. Right panel, pore diameter at V2476. (**B**) A hypothetical mechanistic model for Piezo1 inactivation at the hydrophobic gate in the inner helix. Inactivation is proposed to involve a combined twisting and constricting motion of the inner helix (black arrows), allowing both V2476 and L2475 residues to face the pore to form a hydrophobic barrier.

DOI: https://doi.org/10.7554/eLife.44003.016

Piezo1. One or both of the MF and PE constrictions evident in the cryo-EM structures could conceivably contribute to an activation mechanism, but this remains to be investigated.

The separation of functional gates in Piezo1 is reminiscent of voltage-gated sodium channels (Na$_v$), in which the activation gate is formed by a transmembrane helix, whereas the inactivation gate is formed by an intracellular III-IV linker known as the inactivation ball. This 'ball-and-chain' inactivation mechanism in Na$_v$ channels has been well documented to involve pore block by the inactivation ball (*Shen et al., 2017*; *Yan et al., 2017*; *McPhee et al., 1994*; *West et al., 1992*). However, our data suggest that inactivation in Piezo1 is predominantly accomplished by pore closure via a hydrophobic gate formed by the pore-lining inner helix (*Figure 4A and B*). The proposed inactivation mechanism is also different from that in acid-sensing ion channels (ASICs), in which aspartic acid and glycine residues in a pore-lining helix serve as both an activation and inactivation gate by physically occluding the pore (*Yoder et al., 2018*).

The inactivation rate of Piezo1 channels is voltage modulated (*Coste et al., 2010*; *Moroni et al., 2018*) and depends on a single positively charged K2479 residue in the inner helix (*Wu et al., 2017b*). The putative hydrophobic inactivation gate (L2475/V2476) identified in this study is located just one alpha turn upstream from K2479. The close proximity between these elements suggests there may be functional coupling between the voltage-sensing and inactivation processes, but the exact mechanism remains to be determined. Even though we did not detected a change in the slope of voltage dependence of inactivation between wild type Piezo1 and serine mutations at L2475 and V2476 sites (*Figure 2H*), there remains a possibility that these mutations could affect voltage sensitivity in the range beyond that used in our study.

By combining mutations in the putative hydrophobic inactivation gate and the MF constriction in the CTD, we were able to completely abolish Piezo1 inactivation. These results suggest that the MF constriction plays a minor role in inactivation by acting as a secondary inactivation gate. Indeed, the kinetics of Piezo1 recovery from inactivation strongly suggest the existence of two inactivated states

in the channel (*Lewis et al., 2017*). Further experiments are needed to establish whether the two inactivated states are associated with the two putative gates proposed in this study. A complete elimination of Piezo1 inactivation shows that the two gates are sufficient to account for the full inactivation process in Piezo1. Having two inactivation gates may provide additional dimensions to the regulation of Piezo1 activity. Interestingly, whereas the inner helix site modulates inactivation in both Piezo1 and Piezo2, mutations at the MF constriction only affect Piezo1. Thus, while the two channels share some gating elements, they may not have identical inactivation mechanisms, warranting further studies specifically in Piezo2.

The extracellular cap domain, which is located just above IH, has been shown to be an important modulator of Piezo1 and Piezo2 inactivation. Transposition of the cap domain between the two channels changes inactivation kinetics accordingly (*Wu et al., 2017b*). In the context of our data, it could be that the cap domain acts as a coupling element between force-sensing elements of the channel and the inactivation gate in IH. Understanding the interaction between the cap and IH is important, as these domains carry many disease-associated mutations (*Alper, 2017*; *Wu et al., 2017a*). Even though the LV and MF sites are remarkably conserved among Piezo orthologues, the channels can exhibit prolonged inactivation, as reported for Piezo1 in mouse embryonic stem cells (*Del Mármol et al., 2018*) or Piezo2 in mechanoreceptors from tactile specialist ducks (*Schneider et al., 2017*). In these cases, the slowing of inactivation is probably dictated by other channel regions, post-translational modifications, interaction with regulatory proteins or lipids, which remain to be determined. The three recent cryo-EM structures of Piezo1 are assumed to be in a closed conformation (*Zhao et al., 2018*; *Saotome et al., 2018*; *Guo and MacKinnon, 2017*). To fully understand the conformational changes associated with Piezo1 inactivation, it will be critical to capture Piezo1 and Piezo2 structures in activated and inactivated states, which will provide clues for the development of treatments for Piezo-associated human diseases.

## Materials and methods

**Key resources table**

| Reagent type (species) or resource | Designation | Source or reference | Identifiers | Additional information |
|---|---|---|---|---|
| Cell line (*H. sapiens*) | HEK293T$^{PIEZO1-/-}$ (HEK293T$^{\Delta P1}$) | Dr. Ardem Patapoutian (Scripps Research Institute) (*Lukacs et al., 2015*) | | |
| Recombinant DNA reagent | Mouse-Piezo2-Sport6 | Dr. Ardem Patapoutian (Scripps Research Institute) (*Coste et al., 2010*) | | |
| Recombinant DNA reagent | Mouse-Piezo1-pMO | (*Anderson et al., 2018*) | | |
| Recombinant DNA reagent | Mouse-Piezo1 -IRES-EGFP | Dr. Ardem Patapoutian (Scripps Research Institute) (*Coste et al., 2010*) Addgene #80925 | | |
| Software, algorithm | GraphPad Prism | GraphPad Prism (https://graphpad.com) | RRID:SCR_000306 | Version 7 |
| Software, algorithm | pCLAMP | Molecular Devices (https://www.moleculardevices.com/) | RRID:SCR_011323 | Version 10 |

### Contact for reagent and resource sharing

Further information and requests for resources and reagents should be directed to Sviatoslav Bagriantsev (slav.bagriantsev@yale.edu).

### cDNA constructs and mutagenesis

The Mouse-Piezo2-Sport6 and Mouse-Piezo1-IRES-EGFP (Addgene #80925) were kind gifts from Ardem Patapoutian (Scripps Research Institute, CA) (*Coste et al., 2010*). The Mouse-Piezo1-pMO construct was described elsewhere (*Anderson et al., 2018*). Mutagenesis was performed using the

QuikChange II XL Site-Directed Mutagenesis Kit (Agilent Technologies, La Jolla, CA) and confirmed by sequencing. Mutations in Piezo1 were made in Mouse-Piezo1-pMO except for mutants shown in *Figure 4A*, which were made in Mouse-Piezo1-IRES-EGFP.

## Cell culture and transfection

HEK293T cells with genomic deletion of *PIEZO1* (HEK293T$^{\Delta P1}$, tested negative for mycoplasma) were a kind gift by Ardem Patapoutian (Scripps Research Institute), and were authenticated by PCR and sequencing as described elsewhere (*Lukacs et al., 2015*) (*Dubin et al., 2017*). Cells were cultured in Dulbecco's modified Eagle's medium supplemented with 10% fetal bovine serum and 1% penicillin/streptomycin (ThermoFisher Scientific, Waltham, MA). Transient transfection was performed using Lipofectamine 3000 (ThermoFisher) for Piezo1 or Lipofectamine 2000 (ThermoFisher) for Piezo2 according to the manufacturer's instructions.

## Electrophysiology

Whole-cell patch-clamp recordings of mechano-activated currents from Piezo1 and Piezo2 were performed as previously described (*Anderson et al., 2018*). HEK293T$^{\Delta P1}$ cells transfected with Piezo1 or Piezo2 were seeded onto matrigel-coated coverslips (BD Bioscience, Billerica, MA) 12–48 hr following transfection. For Piezo1 mutants that exhibit dramatically prolonged inactivation, such as L2475Q, V2475Q, LV/QQ or LV/QQ-MF/QQ, current measurements were performed 12–20 hr after transfection. Longer expression times caused toxicity. The extracellular solution contained (in mM): 140 NaCl, 5 KCl, 10 HEPES, 2.5 CaCl$_2$, 1 MgCl$_2$, 10 glucose (pH 7.4 adjusted with NaOH). Recording pipettes were made from borosilicate glass with 1.5 mm outer diameter (Warner Instruments, Hamden, CT) using a micropipette puller (Sutter Instruments, Novato, CA, model P-1000) and polisher (ALA Scientific Instruments, Farmingdale, NY). The polished pipette was back-filled with internal solution containing (in mM): 133 CsCl, 5 EGTA, 1 CaCl$_2$, 1 MgCl$_2$, 10 HEPES, 4 Mg-ATP, 0.4 Na$_2$-GTP (pH 7.3 adjusted with CsOH). The pipette resistance varied from 1 to 3 MΩ when filled with the internal solution. The offset potential was corrected just before the gigaohm seal formation. Series resistance and membrane capacitance were compensated at 85%. Currents were recorded using a Multi-clamp 700-B patch-clamp amplifier and Digidata 1500 digitizer (Molecular Devices, Union City, CA), filtered at 10 kHz through an internal Bessel filter, and sampled at 20 kHz using a 500 MΩ feedback resistor. The pClamp 10 software (Axon Instruments, Union City, CA) was used for data acquisition and analysis. Recordings were not corrected for liquid junction potential.

For whole-cell recordings, mechanical stimuli were applied with a fire-polished, blunt glass pipette (tip diameter,~2–4 μm) controlled by a pre-loaded Piezo actuator stack (Physik Instrumente, Karlsruhe, Germany). After break-in, the tip of the glass probe was positioned just above the cell membrane. The probe was advanced at 1000 μm/s in 1 μm increments at an angle of 30° to the horizontal plane. Cells were held at −80 mV during recordings. The time constant of inactivation ($\tau_{inact}$) was determined by fitting the current decay (between the peak point and the stimulus offset) to a single exponential function: $I = \Delta I * exp(-t/\tau_{inact})$, where $\Delta I$ is the difference between the peak current and baseline, $t$ is the time from the peak current, and $\tau_{inact}$ is the inactivation constant. The apparent threshold of mechano-activated current was defined as the first indentation depth that elicit a peak current greater than background noise signal, typically at least 40 pA.

For cell-attached recordings of mechanically activated Piezo1 current, HEK293T$^{\Delta P1}$ cells were prepared similarly to whole-cell recordings. Fire-polished patch pipettes with resistance of 1–2 MΩ were filled with solution containing (in mM): 130 NaCl, 5 KCl, 10 HEPES, 10 TEA-Cl, 1 CaCl$_2$, 1 MgCl$_2$, pH 7.3 (with NaOH). External solution contained (in mM): 140 KCl, 10 HEPES, 1 MgCl$_2$, 10 glucose, pH 7.3 (with KOH). Stretch-activated Piezo1 currents were stimulated with stepwise, 500 ms negative pressure pulses (Δ10 mmHg with 3 s between stimuli) using a high speed pressure clamp system (HSPC-1, ALA Scientific Instruments). The membrane potential inside the patch was held at −80 mV. Data were recorded at a sampling frequency of 10 kHz using a 5 GΩ feedback resistor.

## Statistical analysis

Data were analyzed and plotted using GraphPad Prism 7.01 (GraphPad Software Inc, La Jolla, CA) and expressed as means ± SEM. Statistical analyses were carried out using Student's tests when comparing two groups or one-way or two-way ANOVA for three or more groups, with corrections

for multiple comparisons. Statistical tests were chosen based on sample size and normality of distribution. Sample size and statistical tests are reported in figure legends. A probability value (p) of less than 0.05, 0.01, 0.001 was considered statistically significant and indicated by *, **, and ***, respectively.

## Acknowledgements

We thank members of the Bagriantsev and Gracheva laboratories for their contributions throughout the project, Evan Anderson for help with high-speed pressure clamp experiments, and Jon Matson for cloning. WZ was supported by James Hudson Brown-Alexander B Coxe Postdoctoral Fellowship. This study was partly funded by NIH grant 1R01NS091300-01A1 (to EOG) and by NSF CAREER grant 1453167 and NIH NINDS grant 1R01NS097547-01A1 (to SNB).

## Additional information

### Funding

| Funder | Grant reference number | Author |
| --- | --- | --- |
| National Institutes of Health | 1R01NS097547-01A1 | Sviatoslav N Bagriantsev |
| National Science Foundation | 1453167 | Sviatoslav N Bagriantsev |
| National Institutes of Health | 1R01NS091300-01A1 | Elena O Gracheva |
| Yale School of Medicine | James Hudson Brown-Alexander B Coxe Fellowship | Wang Zheng |

The funders had no role in study design, data collection and interpretation, or the decision to submit the work for publication.

### Author contributions

Wang Zheng, Conceptualization, Data curation, Formal analysis, Funding acquisition, Investigation, Methodology, Writing—original draft, Writing—review and editing; Elena O Gracheva, Sviatoslav N Bagriantsev, Conceptualization, Supervision, Funding acquisition, Methodology, Writing—original draft, Project administration, Writing—review and editing

### Author ORCIDs

Wang Zheng  http://orcid.org/0000-0001-7577-456X
Sviatoslav N Bagriantsev  http://orcid.org/0000-0002-6661-3403

### Decision letter and Author response

Decision letter https://doi.org/10.7554/eLife.44003.019
Author response https://doi.org/10.7554/eLife.44003.020

## Additional files

### Supplementary files

• Transparent reporting form
DOI: https://doi.org/10.7554/eLife.44003.017

### Data availability

All data generated or analysed during this study are included in the manuscript and supporting files.

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
