## [Decision Letter]

Thank you for submitting your article "A hydrophobic gate in the inner pore helix is the major determinant of inactivation in mechanosensitive Piezo channels" for consideration by *eLife*. Your article has been reviewed by three peer reviewers, including Leon D Islas as the Reviewing Editor and Reviewer #1, and the evaluation has been overseen by Richard Aldrich as the Senior Editor. The following individual involved in review of your submission has also agreed to reveal their identity: Alexander Sobolevsky (Reviewer #3).

The reviewers have discussed the reviews with one another and the Reviewing Editor has drafted this decision to help you prepare a revised submission.

This manuscript presents experiments that identify a region of hydrophobic aminoacids in Piezo1 and Piezo2 channels that is responsible for the inactivation of currents elicited by mechanical stimulation that is a hallmark of these channels. This is a significant advance in understanding the molecular mechanisms of Piezo channels. The three reviewers agree that the work is of high quality and significance and have only a few comments and suggested revisions which are appended below.

Essential revisions:

1) The authors mention that inactivation in Piezo is important for sensory perception in a few specialized organisms. Could you comment if a phylogenetic analysis of this region shows any variability that is consistent with the findings presented? In fact such an analysis would be a nice addition to this manuscript.

2) It is mentioned that hydrophobic gating has been found in TRP channels but this is a generalization. TRPV1 and other TRPV channels have gating mechanisms more akin to Kv channels (Salazar et al. 2009,NSMB).

3) While the manuscript provides important insight into the molecular mechanism of inactivation, the biophysics of this process is glossed over. A main finding is that mutation in the LV site combined with one of the gating constrictions completely eliminates inactivation. This argues that both activation and inactivation might be coupled as in N-type inactivation of Kv channels. Is there any evidence that this might be the case?

4) In Figure 1—figure supplement 1, it may be helpful to quantify deactivation kinetics and include statistics for the measurement.

5) The use of MA current rise time, as in Figure 2E, to assess changes in activation kinetics is limited by the speed of the poke stimulus. The authors may want to comment on this limitation in the Discussion.

6) The LV mutations are located one helical turn above the previously identified voltage-sensing residue K2479. Although the slope sensitivity of the mutants is similar to mP1, as shown in Figure 2H, the possibility remains that the mutants shift the V50 such that the observed slope sensitivity over the tested range remains unchanged. The authors may want to comment on this possibility in the Discussion.

7) Introduction, second paragraph. The structures were not solved to the atomic resolution, so precision of pore diameter estimation cannot possibly be in 1/100 of Å.

8) Subsection “Physical constrictions in the CTD play only moderate roles in Piezo1 inactivation”, first paragraph. In Figure 1D, it would be nice to show control responses of non-transfected HEK293TDeltaP1 cells to MA. My guess is such responses are shown in Figure 2B and can simply be moved to Figure 1D.

9) Subsection “The pore-lining inner helix plays a major role in Piezo1 inactivation”. "MA" should be removed from the first paragraph.

10) Subsection “Mutation of the inner helix and MF constriction eliminates Piezo1 inactivation”. At this point, it is not clear why the MF constriction and LV gate are called secondary and primary gates, respectively. It might have sense to introduce these terms later in the manuscript, after the results for Piezo2 are already presented.

11) Figure 1B-C. To avoid possible confusion, it might be better to have the same coloring of the structure (domain- rather than subunit-based) as in Figure 1A.

12) Figure 4C. Description of the left panel is missing in the figure legend.

---

## [Author Response]

Essential revisions:1) The authors mention that inactivation in Piezo is important for sensory perception in a few specialized organisms. Could you comment if a phylogenetic analysis of this region shows any variability that is consistent with the findings presented? In fact such an analysis would be a nice addition to this manuscript.

We have added additional species to our phylogenetic comparison of the gate regions for Piezo 1 (Figure 2A) and Piezo2 (Figure 5A), for a total of eight species for each channel. As seen in those images, the gates are highly conserved and cannot account for the prolonged kinetics of inactivation observed in Piezo1 in stem cells (del Marmol et al., 2018) or Piezo2 from tactile specialist ducks (Schneider et al., 2017). In these cases, the slowing of inactivation is probably dictated by other channel regions, post-translational modifications, interaction with regulatory proteins or lipids.

We have added the following text:

“Even though the LV and MF sites are remarkably conserved among Piezo orthologues, the channels can exhibit prolonged inactivation, as reported for Piezo1 in mouse embryonic stem cells (del Marmol et al., 2018) or Piezo2 in mechanoreceptors from tactile specialist ducks (Schneider et al., PNAS 2017). In these cases, the slowing of inactivation is probably dictated by other channel regions, post-translational modifications, interaction with regulatory proteins or lipids, which remain to be determined.”

2) It is mentioned that hydrophobic gating has been found in TRP channels but this is a generalization. TRPV1 and other TRPV channels have gating mechanisms more akin to Kv channels (Salazar et al. 2009,NSMB).

We agree with the comment and have modified the text as follows:

“For example, even though a number of TRP channels, including TRPV1, have a gating mechanism similar to that found in voltage-gated potassium channels (Salazar et al., 2009), others, such as TRPC2 and TRPP2 contain a hydrophobic activation gate in the cytoplasmic pore-lining S6 helix, which was revealed by both electrophysiological (Zheng et al., 2018b; Zheng et al., 2018a) and structural studies (Cheng, 2018).”

3) While the manuscript provides important insight into the molecular mechanism of inactivation, the biophysics of this process is glossed over. A main finding is that mutation in the LV site combined with one of the gating constrictions completely eliminates inactivation. This argues that both activation and inactivation might be coupled as in N-type inactivation of Kv channels. Is there any evidence that this might be the case?

This is a very interesting point. Our data suggest that activation and inactivation of Piezos are carried out by separate channel elements, and in this sense the mechanism is similar to that found in Kvs. However, inactivation of Piezos occurs at the LV and MF sites along the pore, and which is mechanistically different from the N-type mechanism in Kvs. At this point, we do not have evidence to suggest that activation and inactivation in Piezos is coupled. This is an important topic that necessitates further investigation.

4) In Figure 1—figure supplement 1, it may be helpful to quantify deactivation kinetics and include statistics for the measurement.

The quantification and statistical analysis of deactivation kinetics has now been included in Figure 1—figure supplement 1.

5) The use of MA current rise time, as in Figure 2E, to assess changes in activation kinetics is limited by the speed of the poke stimulus. The authors may want to comment on this limitation in the Discussion.

We have now acknowledged this experimental limitation as follows:

“We also did not detect a change in MA and current rise time, even though a small change could avoid detection due to limitations imposed by the velocity of the mechanical probe.”

6) The LV mutations are located one helical turn above the previously identified voltage-sensing residue K2479. Although the slope sensitivity of the mutants is similar to mP1, as shown in Figure 2H, the possibility remains that the mutants shift the V50 such that the observed slope sensitivity over the tested range remains unchanged. The authors may want to comment on this possibility in the Discussion.

We have now acknowledged this possibility as follows:

“Even though we did not detected a change in the slope of voltage dependence of inactivation between wild type Piezo1 and serine mutations at L2475 and V2476 sites (Figure 2H), there remains a possibility that these mutations could affect voltage sensitivity in the range beyond that used in our study.”

7) Introduction, second paragraph. The structures were not solved to the atomic resolution, so precision of pore diameter estimation cannot possibly be in 1/100 of Å.

We have changed the values of pore diameter accordingly, both in text and in Figure 1C:

“These constrictions define minimum pore diameters of 6 Å and 4 Å, respectively, thus the structures are assumed to represent a closed state.”

8) Subsection “Physical constrictions in the CTD play only moderate roles in Piezo1 inactivation”, first paragraph. In Figure 1D, it would be nice to show control responses of non-transfected HEK293TDeltaP1 cells to MA. My guess is such responses are shown in Figure 2B and can simply be moved to Figure 1D.

The control response has been moved from Figure 2B to Figure 1D, as suggested.

9) Subsection “The pore-lining inner helix plays a major role in Piezo1 inactivation”. "MA" should be removed from the first paragraph.

Removed.

10) Subsection “Mutation of the inner helix and MF constriction eliminates Piezo1 inactivation”. At this point, it is not clear why the MF constriction and LV gate are called secondary and primary gates, respectively. It might have sense to introduce these terms later in the manuscript, after the results for Piezo2 are already presented.

We agree that using the terms “primary” and “secondary” is premature at this point. We have modified this sentence as follows:

“Thus, our data suggest that the MF constriction in the CTD could act in concert with the inner helix hydrophobic LV gate to produce fast inactivation of Piezo1.”

We introduce the terms “primary” and “secondary” inactivation gates later in discussion:

“Collectively, our data lead us to conclude that the two residues at the LV site form a hydrophobic inactivation gate, which contributes to the majority of MA current decay (primary inactivation gate), and that the MF constriction acts as a secondary inactivation gate in Piezo1.”

11) Figure 1B-C. To avoid possible confusion, it might be better to have the same coloring of the structure (domain- rather than subunit-based) as in Figure 1A.

The coloring has been unified, as suggested.

12) Figure 4C. Description of the left panel is missing in the figure legend.

We’ve corrected the description to Figure 1C:

“(C) Representative cell-attached MA current traces induced by high-speed pressure clamp via application of a negative pipette pressure in HEK293T^ΔP1^ cells expressing GFP (negative control), WT or mutant Piezo1. E_hold_ = 80 mV.”